# Excitonic insulator to superconductor phase transition in ultra-compressed helium

Cong Liu[1], Ion Errea [2,3,4], Chi Ding[5], Chris Pickard [6,7], Lewis J. Conway[6,7], Bartomeu Monserrat [6,8], Yue-Wen Fang [2,3], Qing Lu[5], Jian Sun [5] ✉, Jordi Boronat [1] & Claudio Cazorla [1] ✉

Helium, the second most abundant element in the universe, exhibits an extremely large electronic band gap of about 20 eV at ambient pressures. While the metallization pressure of helium has been accurately determined, thus far little attention has been paid to the specific mechanisms driving the band-gap closure and electronic properties of this quantum crystal in the terapascal regime (1 TPa = 10 Mbar). Here, we employ density functional theory and many-body perturbation calculations to fill up this knowledge gap. It is found that prior to reaching metallicity helium becomes an excitonic insulator (EI), an exotic state of matter in which electrostatically bound electron-hole pairs may form spontaneously. Furthermore, we predict metallic helium to be a superconductor with a critical temperature of ≈ 20 K just above its metallization pressure and of ≈ 70 K at 100 TPa. These unforeseen phenomena may be critical for improving our fundamental understanding and modeling of celestial bodies.

In their final evolution stage, the majority of stars in the Universe become white dwarfs (WDs) mostly consisting of a mixture of helium, carbon, and oxygen atoms immersed in a sea of electrons. The lower the mass of the WD, the larger the relative abundance of helium. Due to the lack of a continuous energy source, WDs cool down over time eventually reaching a temperature of 2.7 K (i.e., the current temperature of the Universe due to cosmic background radiation)[1–3]. Understanding the cooling process of WDs is essential to infer their progression over time and thus provide reliable bounds for the age of the Universe. In the interior of WDs, pressure may reach values billions of times higher than that in the Earth's surface (10–100 TPa), which at present are not attainable in experiments. (Current laser-driven ramp compression experiments at the National Ignition Facility, for example, reach top pressures of about 5 TPa[4–6]). Consequently, theoretical modeling of light materials under extreme compression conditions,

and in particular of helium, turns out to be critical for probing the interior of WDs and comprehend their physico-chemical evolution.

Highly accurate diffusion Monte Carlo (DMC) calculations predict solid helium to become a metal in the hexagonal closed packed (hcp) phase at a pressure of 25.7 TPa[7]. By considering zero-point energy and electron-phonon coupling effects estimated with density functional theory (DFT) methods, such a metallization pressure increases up to 32.9 TPa at $T = 0$ K[8]. (At much higher temperatures, in the fluid phase, DFT calculations provide smaller metallization pressures of the order of 1–10 TPa[9]). Both experimental and theoretical studies have shown that the valence-band maximum (VBM) in solid ⁴He appears on the line joining the reciprocal lattice points Γ (0, 0, 0) and M (q, 0, 0), while the conduction-band minimum (CBM) is located at the Γ point[8,10]. Thus, the band gap of solid helium is indirect and according to previous DFT calculations the overlap between the conduction and valence bands

---

[1]Departament de Física, Universitat Politècnica de Catalunya, Campus Nord B4-B5, Barcelona 08034, Spain. [2]Fisika Aplikatua Saila, Gipuzkoako Ingeniaritza Eskola, University of the Basque Country (UPV/EHU), Europa Plaza 1, 20018 Donostia/San Sebastián, Spain. [3]Centro de Física de Materiales (CSIC-UPV/EHU), Manuel de Lardizabal pasealekua 5, 20018 Donostia/San Sebastián, Spain. [4]Donostia International Physics Center (DIPC), Manuel de Lardizabal pasealekua 4, 20018 Donostia/San Sebastián, Spain. [5]National Laboratory of Solid State Microstructures, School of Physics and Collaborative Innovation Center of Advanced Microstructures, Nanjing University, Nanjing 210093, China. [6]Department of Materials Science and Metallurgy, University of Cambridge, Cambridge CB30FS, UK. [7]Advanced Institute for Materials Research, Tohoku University, Sendai 980-8577, Japan. [8]Cavendish Laboratory, University of Cambridge, Cambridge CB30HE, UK. ✉e-mail: jiansun@nju.edu.cn; claudio.cazorla@upc.edu

upon its closure is characteristic of a semimetal (i.e., the density of electronic states at the Fermi level is very small)[8]. Meanwhile, the lattice phonons involving atomic displacements perpendicular to the hcp basal plane drive the widening of the band gap at very high pressures[8].

A detailed understanding of the electronic band structure properties of this archetypal quantum crystal[11], however, is still lacking. First, about half a century ago the existence of an exotic insulating phase called "excitonic insulator" (EI) was predicted in which electrons and holes spontaneously form bound pairs called excitons[12]. The EI phase could be stabilized at sufficiently low temperatures in semiconductors with tiny band gaps or semimetals with very small band overlaps. Recently, experimental EI fingerprints have been reported for low-dimensional transition metal dichalcogenide structures exhibiting small band gaps[13,14]; however, stabilization of a bulk EI state remains elusive. Owing to its semiconductor nature, absence of structural transformations and marked quantum character, ultra-compressed $^4$He appears to be an excellent candidate in which a bulk EI state could emerge and genuine quantum many-body phenomena like high-temperature excitonic superconductivity and BEC–BCS cross-over might exist[15,16]. Is possibly solid helium a bulk EI in the TPa regime? And secondly, the substantial electron-phonon coupling and semimetal Fermi surface previously disclosed in solid helium suggest the possibility of superconductivity in this quantum crystal upon band-gap closure. Is metallic helium a superconductor? If so, what are the underlying physical mechanisms and corresponding critical temperature? Besides their fundamental interest, answering to these questions may have major consequences in the fields of planetary science and astrophysics since this new knowledge could improve our understanding of the thermal and chemical evolution of low-mass WDs[1,2].

In this study, we employ theoretical first-principles approaches based on DFT and many-body perturbation GW calculations to advance knowledge on the electronic, elastic and superconductor properties of solid helium in the TPa regime. Our main finding is an unprecedented bulk excitonic insulator to superconductor phase transition driven by pressure in which the superconductor state can reach a critical temperature of ≈ 70 K under a compression of 100 TPa. It is worth noting that an exhaustive random sampling of the structural space of solid helium was performed at $P = 100$ TPa (AIRSS[17,18]), with the finding that the hcp phase imperturbably remains the ground state (Methods). (Despite the fact that in the TPa regime the $c/a$ ratio of this phase significantly departs from the ideal value of 1.63, we keep labeling it as hcp since its crystal symmetry does not vary.)

## Results and discussion

We started by benchmarking different families of DFT functionals (i.e., semi-local, van der Waals corrected and hybrid)[11] against the metallization pressure of solid helium calculated with DMC methods, which amounts to 25.7 TPa (Fig. 1a). In all the analyzed cases, the band gap decreases almost linearly under increasing pressure due to the steady enhancement of electronic delocalization among neighboring atoms (Fig. 1b). The semi-local PBE functional predicts a metallization pressure of 17 TPa, in consistent agreement with previous computational studies[7,8]. Meanwhile, the hybrid functional B3LYP performs the best in comparison to the DMC benchmark by providing a metallization pressure of ≈ 23 TPa (Fig. 1a and Fig. S1). Van der Waals corrections turn out to be practically negligible in the TPa regime (e.g., the two PBE and PBE-D3 curves practically overlap each with the other) due to the dominant role of interatomic repulsive interactions at short distances[19]. Based on these results, we adopted the hybrid functional B3LYP for our subsequent analysis of the electronic band structure of solid helium.

Unlike atomic hydrogen, hcp $^4$He presents an indirect band gap with the VBM located at the reciprocal point Λ and the CBM at the center of the Brillouin zone (Γ point, Fig. 1c), in consistent agreement

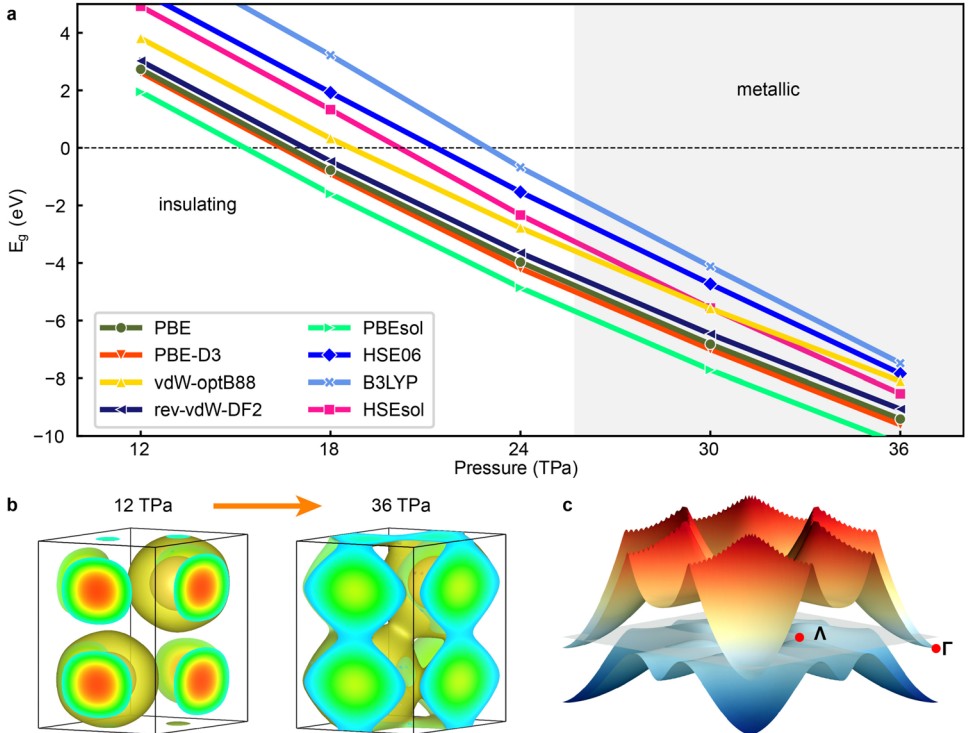

**Fig. 1 | DFT benchmarking for the metallization pressure of hcp $^4$He. a** Electronic band gap, $E_g$, expressed as a function of pressure and calculated with different DFT functionals. Negative $E_g$ values indicate overlapping between the VBM and CBM levels. The grey region indicates the stability range of metallic helium as calculated with QMC methods[7]. **b** Electronic localization function (ELF isosurface = 0.8, yellow) of solid helium at 12 and 36 TPa in a red-green-blue color scale with red denoting high electronic density and blue low electronic density. **c** Lowest conduction (red) and highest valence (blue) bands expressed as a function of reciprocal wave vector in the $k_z = 0$ plane; the grey and transparent plane represents the Fermi surface.

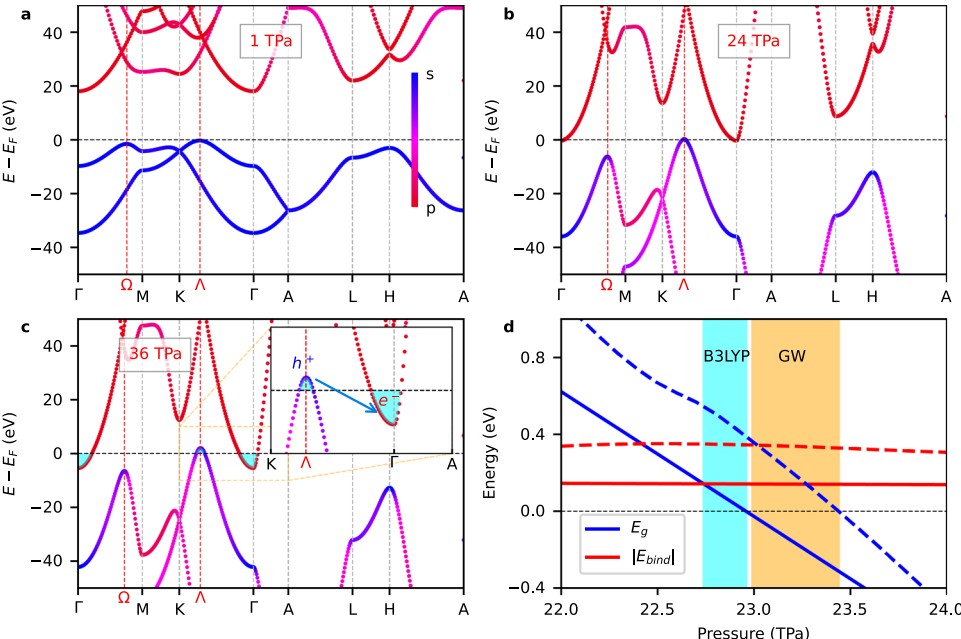

**Fig. 2 | Band-gap closure and emergence of the excitonic insulator state in hcp ⁴He. a–c** Evolution of the electronic band structure under compression calculated with the hybrid B3LYP functional. A red-magenta-blue color scale is employed for representing the orbital character of the relevant electronic bands with blue for $s$- and red for $p$-like. The $\Lambda$ and $\Omega$ reciprocal space points indicate the location of the primary and secondary VBM. The inset highlights the migration of electrons ($e^-$) and formation of holes ($h^+$) along the reciprocal space line K-$\Lambda$-$\Gamma$ near the Fermi surface. **d** Comparison of the excitonic binding energy, $|E_{bind}|$, and band gap calculated with B3LYP DFT (solid lines) and many-body perturbation theory within the GW approximation (dashed lines). The cyan (B3LYP) and orange (GW) regions indicate the pressure range in which the sufficient condition for spontaneous formation of excitons is fulfilled.

with previous DFT calculations and experiments[8,10]. It is noted that the direct band gap at the $\Lambda$ point actually increases under compression (Fig. S2). Interestingly, when the energy gap between the VBM and CBM levels disappears a semimetal state characterized by an almost negligible density of states at the Fermi level emerges due to the fact that no additional electronic bands cross the Fermi surface (Fig. 2a–c). At 1 TPa, the VBM consists exclusively of $s$-like orbitals while the CBM exhibits full $p$-like character (Fig. 2a). Upon further compression, the VBM presents increasingly larger hybridization between $s$ and $p$-like orbitals while the CBM conserves its pure $p$-like character (Fig. 2b). At pressures higher than 23 TPa, electrons from the VBM at the $\Lambda$ point are transferred to the CBM at $\Gamma$ in order to lower their energy, thus rendering a $p$-type semimetal system (Fig. 2c).

The continuous pressure-driven closure of the band gap and subsequent stabilization of a semimetal state in hcp ⁴He, suggest the possibility of spontaneous formation of excitons with finite momentum $|q| = \Lambda-\Gamma$ at low temperatures. An exciton is a bound state formed by an excited electron ($e^-$) in the conduction band and a hole ($h^+$) in the valence band that interact through attractive Coulomb forces. In narrow-gap semiconductors, a sufficient condition for the spontaneous formation of excitons is that the corresponding binding energy, $E_{bind}$, is larger in absolute value than the band gap since then the total energy of the system can be lowered by promoting electrons to the conduction band in the absence of optical excitations[16,20]. We computed the binding energy of an exciton in ultra-compressed hcp ⁴He by relying on the Wannier-Mott model ("Methods") since the dielectric constant of solid helium in the TPa regime is relatively high ($\epsilon_r > 5$, Fig. S3) and consequently electric field screening effects are large[21].

Our excitonic binding energy results obtained with the hybrid B3LYP functional and expressed as a function of pressure are shown in Fig. 2d. It was found that the sufficient condition for spontaneous formation of excitons, namely, $|E_{bind}| > E_g$, was fulfilled over a wide pressure interval of approximately 0.2 TPa prior to metallization (cyan region, in which $|E_{bind}|^{B3LYP} = 0.19$ eV). In view of this result, we

performed many-body perturbation theory calculations within the GW approximation to explicitly and more accurately determine quasiparticle excitations in ultracompressed ⁴He ("Methods")[22]. As it is shown in the inset of Fig. 2d, GW calculations provided a much larger excitonic binding energy than calculated with the Wannier-Mott model and hybrid DFT functionals, namely, $|E_{bind}|^{GW} = 0.34$ eV. Moreover, the estimated pressure interval in which excitons can spontaneously form noticeably increased up to 0.45 TPa (orange region). Therefore, based in our hybrid DFT and many-body perturbation GW calculations we may conclude that on the verge of metallization hcp ⁴He is a bulk excitonic insulator (EI). The same conclusion was reached when considering alternative structural phases for ultra-compressed solid helium along with semi-local DFT functionals (Fig. S4).

The emergence of a bulk EI state is expected to be accompanied by strong lattice distortions and instabilities due to arising electron-phonon interactions[23,24]. We computed the phonon spectrum of hcp ⁴He at different pressures using the semi-local PBE and PBEsol functionals (since phonon calculations at this level of theory are feasible), as shown in Figs. 3a and Fig. S5. (The hybrid B3LYP functional certainly provides very similar phonon spectrum results than obtained with semi-local DFT functionals when employing small supercells, Fig. S6.) Reassuringly, a distinct phonon softening appears at the reciprocal lattice point $\Lambda$ between 15 and 20 TPa, that is, when semi-local DFT functionals predict that solid helium becomes a metal (Fig. S7). Interestingly, above 30 TPa additional phonon softenings emerge along the K-$\Gamma$ and M-K reciprocal space directions; we found that around this pressure the energy gap between the CBM (located at $\Gamma$) and secondary VBM (located at $\Omega$, Figs. 2c and Fig. S7) vanished. Thus, in addition to validating our prediction for the stabilization of a bulk EI state, these findings corroborate the strong coupling between electrons and lattice vibrations previously disclosed in ultra-compressed solid helium[8]. It is worth noting that quantum anharmonic effects were assessed for ⁴He with the stochastic self-consistent harmonic approximation (SSCHA) method[25–28] and found to be of little relevance in the TPa regime ("Methods" and Fig. S8).

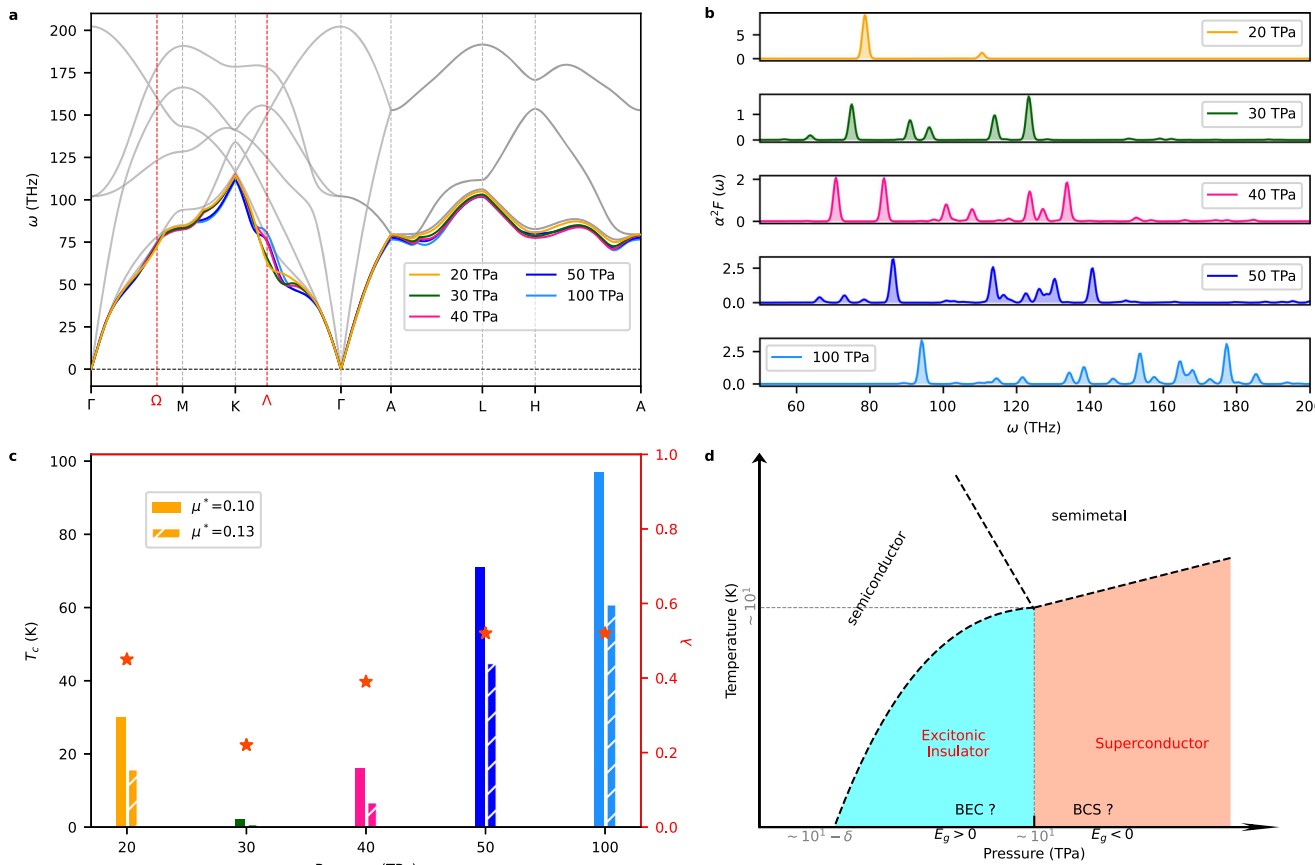

**Fig. 3 | Electron-phonon coupling and superconducting properties of ultra-compressed hcp $^4$He. a** Pressure-induced variation of the acoustic phonon branches (colored lines). The phonon calculations were performed with the semi-local PBE functional. The grey curves were calculated for the insulating phase at 15 TPa; the colored lines correspond to higher pressures but were re-scaled to facilitate the comparison. **b** Eliashberg spectral function, $\alpha^2F$, of metallic hcp $^4$He. **c** Superconducting properties estimated with the semi-local PBE functional and

parameters $\mu^* = 0.10$ and $\mu^* = 0.13$ (Methods). The critical superconducting temperature values, $T_c$ (colored bars), obtained with the modified Allen-Dynes formula[29] and the electron-phonon coupling strength parameter, $\lambda$ (red stars), are represented in the left and right ordinate axis, respectively. **d** Qualitative sketch of the possible phase diagram of ultra-compressed solid helium based on work[16] and the key physical findings presented in this study.

Besides lattice dynamics, the elastic, structural and thermodynamic properties of solid helium were also found to be influenced by the pressure-driven EI to metal phase transition (Fig. S9). In the insulating phase, the elastic constants of hcp $^4$He display a practically linear dependence on pressure whereas in the metallic phase they depart from this behavior and in some cases do not even display a monotonic increase under compression (e.g., $C_{13}$). Similar effects were also observed for the bulk and shear moduli, sound velocities, Debye temperature and heat capacity (Fig. S9). Regarding the structural features, it was found that the pressure evolution of the hcp lattice parameter ratio $c/a$ drastically changes when the metallic phase is stabilized (Fig. S10). It is worth noting, however, that equivalent alterations were not perceptible in the zero-temperature equation of state (Fig. S11). Thus, taking into account these unanticipated structural, elastic and thermodynamic effects could have important consequences on current modeling of astrophysical bodies, in particular, of low-mass and helium-rich WDs[1,2].

Motivated by the findings described above, we explored the superconducting properties of ultra-compressed hcp $^4$He (using the computationally feasible PBE functional). Accurate electron-phonon coupling (EPC) calculations were carried out with the techniques outlined in the Methods section, which essentially involve the Bardeen-Cooper-Schrieffer (BCS) theory of superconductivity and modified Allen-Dynes formula[29]. Figure 3b shows the Eliashberg spectral function, $\alpha^2F$, estimated at different pressures, from which the

corresponding average EPC strength, $\lambda$, can be straightforwardly computed (Methods and Supplementary Information). At 20 TPa, this function exhibits two appreciable peaks: the most prominent appearing at low frequencies stems from the lowest-energy phonon band at the wave-vector $\mathbf{q} = (0.25, 0.43, 0)$, which is very close to the reciprocal space point $\Lambda$ associated with phonon softening; the other peak emerging at higher frequencies stems from the second and third lowest-energy phonon bands at $\Gamma$. The EPC strength associated to these phonon modes respectively amount to ≈140 and 40 (Tables S1–S2), which are extremely high as compared to the average $\lambda$ of the high-temperature superconductors H$_3$S[30] and LaH$_{10}$[31] (i.e., ~1–10). Such giant $\lambda_{qv}$ values result from huge phonon linewidths and the minute density of electronic states at the Fermi level ("Methods", Supplementary Information and Fig. S7). However, since the number of phonon modes that appreciably contribute to $\alpha^2F$ (or, equivalently, to $\lambda$) is quite reduced, the superconducting temperature estimated at 20 TPa is relatively low, namely, $T_c = 17$ K (Fig. 3c and Supplementary Information).

Interestingly, upon further compression, when the energy overlap between the conduction and valence bands is enhanced (Fig. 2c), additional peaks appear in the Eliashberg spectral function that noticeably contribute to the average EPC strength, thus raising the superconducting critical temperature. For instance, at a pressure of 50 TPa, when multiple phonon softenings and $\alpha^2F$ local maxima are observed (Fig. 3a, b), we estimated a substantial superconducting

critical temperature of 47 K (Fig. 3c and Supplementary Information). Under higher compression the superconducting critical temperature steadily increases, reaching a peak value of ≈70 K at the maximum pressure of 100 TPa considered in our calculations (Fig. 3c). It is worth noting that within the pressure interval $20 \leq P \leq 30$ TPa both $T_c$ and $\lambda$ noticeably decrease; this transient effect is due to a dominant $P$-induced surge in the Fermi density of electronic states that drastically reduces the $\alpha^2 F$ peaks (Fig. 3b, "Methods" and Table S3).

An analogous EPC strength parameter and $T_c$ analysis was carried out for hcp xenon (Fig. S12) since this material is isoelectronic to solid helium and becomes metallic at experimentally accessible pressures of the order of 0.1 TPa. An EI to metal phase transition similar to that disclosed in ultra-compressed $^4$He was also found for hcp Xe at 0.14 TPa. A noticeable phonon softening appeared at a higher pressure of 0.19 TPa, coinciding with the closure of a secondary band gap involving a $s$-like dominant CBM and $p$-like dominant VBM (i.e., of the same character than the primary band gap in solid helium). The EPC strength and superconducting critical temperature estimated for hcp Xe at 0.14 TPa are 0.75 and ≈10 K, respectively. Thus, bulk Xe seems to be a good candidate material in which to experimentally search for analogs of some of the key theoretical findings revealed in this work for ultra-compressed hcp $^4$He.

As it was mentioned previously, EPC effects on the atomic positions significantly increase the zero-temperature metallization pressure of solid helium by several TPa[8]. Consequently, a well-founded question is: how such type of EPC effects may affect the excitonic insulator and $T_c$ results presented in this study? To quantitatively answer this question (although approximately, due to the lack of computationally affordable approaches), we proceeded as described next. First, to evaluate the influence of EPC effects on the excitonic binding energy, we generated a number of supercell configurations for a pressure of 18 TPa in which the atoms were displaced along representative thermal lines determined with the method described in work[32]. Subsequently, we calculated $E_{bind}$ for those configurations by using the Wannier-Mott model, along with $E_g$, and computed their average value ("Methods" and Supplementary Information). It was found that EPC effects (1) significantly open the band gap, in agreement with previous DFT calculations[8], and (2) enhance the excitonic binding energy very substantially by >40% ("Methods" and Supplementary Information). And second, we carried out a perturbative $T_c$ calculation in which we combined the phonon spectrum obtained at ≈30 TPa, that is, the pressure at which solid helium becomes metallic when taking into consideration EPC effects, and the electronic band structure obtained at ≈20 TPa by neglecting EPC effects, which renders an incipient metallic state ("Methods" and Supplementary Information). By proceeding so, we obtained a corrected $T_c$ for a nominal pressure of ≈30 TPa that was six times higher than the corresponding value obtained when disregarding dynamical effects. Therefore, we may conclude that EPC on the atomic positions, in addition to displace the metallization of solid $^4$He towards higher compression, is also likely to enhance the excitonic binding energies and superconductor transition temperatures reported in this study.

Figure 3d shows a sketch of the possible phase diagram of solid helium at pressures and temperatures that are relevant to astrophysical studies. At sufficiently high pressures and low temperatures, a bulk EI state is stabilized. Whether in such a state the spontaneously created electron-hole bound pairs form excitonic Bose-Einstein condensates or exhibit excitonic superconductivity close to zero temperature[15,16], is a matter that we cannot resolve with the DFT-based methods employed in this work (hence the questions marks in the figure). Upon further compression, hcp $^4$He becomes a superconductor with a critical temperature that increases under pressure (made the exception of a small pressure interval following metallization, which has been neglected in the figure). At high enough temperatures, superconductor solid helium transforms into a $p$-type

semimetal. These electronic phase transitions significantly impact the structural, elastic, thermodynamic and transport properties of hcp $^4$He hence should be taken into consideration in advanced evolutionary models of stellar bodies like white dwarfs.

In conclusion, we have presented a comprehensive first-principles computational study of the physical properties of solid helium in the TPa regime, putting special emphasis on its electronic band-structure features. It was found that over a broad pressure range preceding metallization hcp $^4$He becomes a bulk excitonic insulator in which electrostatically bound electron-hole pairs can form spontaneously. This bulk excitonic insulator state could host genuine quantum many-body phenomena like high-temperature excitonic superconductivity and excitonic BEC–BCS crossover, although additional advanced studies are necessary to fully assess these hypotheses. Upon band-gap closure, solid helium transitions into a superconductor state that possesses a critical temperature of the order of $10^1$–$10^2$ K, depending on compression. This pressure-induced EI to superconductor phase transition is accompanied by several elastic and structural anomalies. Thus, our theoretical findings besides conveying great fundamental interest are also of great relevance to the physics of celestial bodies, in particular, of low-mass WDs mostly containing metallic helium. Furthermore, it is argued that some analogs of the key theoretical findings revealed here for ultra-compressed helium could be experimentally observed in solid xenon.

## Methods

### First-principles calculations outline

Density functional theory (DFT) calculations were performed with the Vienna ab initio simulation package (VASP)[33]. The projector augmented-wave (PAW) method[34] was employed and the $1s^2$ electrons in the He atoms were treated as valence. Different families of DFT functionals were tested among which we highlight the semi-local Perdew-Burke-Ernzerhof (PBE)[35] and revised PBE for solids (PBEsol)[36], van der Waals corrected DFT-D3[37], non-local dispersion corrected vdW-optB88[38], vdW-DF-cx[39], rev-vdW-DF2[40], and the hybrid HSE06[41], B3LYP[42] and HSEsol[43]. Different functionals provide analogous equations of state (Fig. S11). A plane wave energy cutoff of 1500 eV was employed along with dense Monkhorst-Pack $k$-point sampling grids of resolution $2\pi \times 0.025$ Å (Fig. S13). The energy and atomic forces in the structural relaxations were converged to within $10^{-6}$ eV and 0.002 eV/Å, respectively. For validation purposes, we compared our band gap results obtained with the PBE functional as implemented in the VASP code with a full-potential (linearized) augmented plane-wave method as implemented in the WIEN2k code[44] (Fig. S14). Phonon calculations were performed with the small displacement method and the PHONOPY code[45] by employing large supercells of $4 \times 4 \times 4$. A phonon spectrum test was performed for the hybrid B3LYP functional considering a relatively small $2 \times 2 \times 2$ supercell (due to computational limitations). We found that the phonon spectra obtained with the hybrid B3LYP and semi-local PBE functionals presented only minor differences around the $\Gamma$ point (Fig. S6).

The excitonic binding energy was estimated with the Wannier-Mott formula:

$$E_{bind} = - (m_u{}^* Ry)/(m_0 \epsilon_r^2), \tag{1}$$

where $m_u = (m_e \cdot m_h)/(m_e + m_h)$. In the equation above, $m_e$ and $m_h$ are the effective mass of the electron at the bottom of the conduction band and the hole at the top of the valence band, respectively. $Ry$ represents the Rydberg constant (=13.6 eV), $m_0$ the rest mass of the electron, and $\epsilon_r$ the dielectric constant of the system as referred to vacuum. The electron and hole effective masses were computed like the inverse of the second derivative of the conduction and valence band energies with respect to crystal momentum module, $|k|$, along the reciprocal space path $\Lambda$-$\Gamma$. The Wannier-Mott formula is a good

approximation for the excitonic binding energy of materials possessing high dielectric constants[21], which is the case of hcp $^4$He in the TPa regime (Fig. S3).

The elastic tensor was determined at zero temperature by performing six finite lattice distortions and four atomic displacements of 0.01 Å along each Cartesian direction. The adiabatic bulk modulus, $K$, and shear modulus, $G$, were obtained by computing the Voigt-Reuss-Hill averages from the elastic tensor. The longitudinal and transverse sound velocities were calculated with the formulas $v_p = \left[(K + \frac{4}{3}G)/\rho\right]^{1/2}$ and $v_s = \left[G/\rho\right]^{1/2}$, respectively, where $\rho$ represents the atomic density of the system.

## Crystal structure prediction analysis

The ab initio random structure searching (AIRSS) package[17,18] was used to perform crystal structure searches for solid $^4$He. The first-principles DFT code CASTEP[46] was employed to perform the underlying electronic structure calculations based on the PBE functional[35]. The searches were performed at 100 TPa, producing approximately 1000 relaxed structures and considering a total of 12 atoms in the simulation cell. The energy cutoff was set to 1000 eV and a specially designed hard on-the-fly potential was employed for the calculations. Under these conditions, it was found that the hexagonal $P6_3/mmc$ (hcp) phase remained the ground state followed by a rhombohedral $R\overline{3}m$ phase with a higher relative energy of 0.131 eV/atom. Other energetically competitive structures were an hexagonal $P\overline{6}m_2$ (0.140 eV/atom) and a cubic $Im\overline{3}m$ (0.170 eV/atom) phase.

## SSCHA calculations

Quantum anharmonic effects were assessed with the stochastic self-consistent harmonic approximation (SSCHA) method[25–28]. All the SSCHA calculations were evaluated at the pressure of 35 TPa and 0 K, conditions at which excellent convergence of the SSCHA minimization has been verified by an extra population including 800 supercell configurations. SSCHA calculations were performed with a $6 \times 6 \times 3$ supercell including 216 atoms, which yields the dynamical matrices on a commensurate **q**-mesh of $6 \times 6 \times 3$. The trial harmonic dynamical matrices used for initializing the free energy were obtained from the DFPT method as implemented in the Quantum Espresso (QE) code in the corresponding commensurate **q**-mesh. In the self-consistent calculations of the supercells, we used the same cutoff energy as the electron-phonon coupling calculations for the primitive cell, but the **k**-mesh was reduced accordingly and was tested for convergence. In the SSCHA iterations, except the first four populations in which only internal coordinates were optimized to speed up the minimization, the free energy in other populations was minimized with respect to all degrees of freedom of the crystal structure including the internal coordinates and the lattice cell parameters.

## Many-body perturbation theory calculations within the GW approximation

The excitonic binding energy was also estimated by means of highly accurate many-body perturbation theory calculations within the GW approximation[47] performed with the Yambo code[48]. For this, we employed the generalized gradient approximation (GGA) as parameterized by PBE together with a plane-wave basis set and norm-conserving pseudopotential. The kinetic energy cutoff for the wave functions was set to 600 Ry. The Brillouin zone was sampled with a $64 \times 64 \times 32$ k-mesh. Many-body quasi-particle energies were obtained within the GW approximation[49] by considering 8 $G_0W_0$ iterations, and the dynamic dielectric function was obtained with the full-frequency method with up to 800 integration points. The exciton energies were calculated by solving the Bethe-Salpeter equation[22] with the Tamm-Dancoff approximation[50]. The static screening in the direct term was calculated within the random-phase approximation with the inclusion of local field effects. We used 2 valence and 3 conduction bands to solve the Bethe-Salpeter equation matrix. For the GW band-structure calculations, we sampled the Brillouin zone with a $16 \times 16 \times 8$ **k**-point grid. A kinetic energy cutoff of 90 Ry was used for the evaluation of the exchange part of the self-energy and of 150 Ry for the dielectric screening matrix size. About one hundred unoccupied bands were used to build the polarizability and integrate the self-energy. The convergence test on the number of frequency points and perturbation iterations are shown in Fig. S15. The exciton energies were mapped along high symmetry paths for different pressures, as it is shown in Fig. S16 and summarized in Table S4.

## Electron-phonon coupling calculations and critical superconducting temperature

Electron phonon coupling (EPC) calculations were performed with the Quantum Espresso (QE) code[51,52] by using ultrasoft pseudopotentials (i.e., semi-local PBE[35]), an energy cutoff of 200 Ry for the kinetic energy and an energy cutoff of 2000 Ry for the charge density (convergence tests are shown in Table S1). The equation of state of hcp $^4$He computed with the VASP and QE codes show very good agreement, as it is illustrated in Fig. S10. The electron-phonon matrix elements were calculated in a $16 \times 16 \times 8$ **q**-point grid with density functional perturbation theory (DFPT)[53]. We adopted a dense and shifted $k$-point mesh of $80 \times 80 \times 40$ to increase the convergence in the self-consistent calculations. For the EPC calculations, we further increased the $k$-point mesh up to $192 \times 192 \times 96$ (convergence tests are shown in Table S2) and to ensure $k$-point sampling convergence we employed the Methfessel-Paxton scheme with a smearing width of 0.02 Ry. The Dirac deltas on the band energies were substituted by Gaussian functions with a broadening of 0.002 Ry, which are necessary for the calculation of the EPC strength parameter $\lambda$ (convergence tests are shown in Fig. S17). Convergence tests on the **q**-point grid sampling are also presented in Table S5. For further validation purposes, we also performed calculations with a hardcore pseudopotential involving a real-space cutoff of $r_c = 0.37a_0$[54] and compared the results with those obtained with ultrasoft pseudopotentials, as it is shown in Fig. S18.

The Eliashberg spectral function, $\alpha^2 F(\omega)$, accounts for the coupling between phonons and electrons in the Fermi surface like:

$$\alpha^2 F(\omega) = \frac{1}{2\pi\hbar N(E_F)N_{q\nu}} \sum_{q\nu} \frac{\gamma_{q\nu}}{\omega_{q\nu}} \delta(\omega - \omega_{q\nu}), \qquad (2)$$

where $N(E_F)$ is the density of states at the Fermi level (per unit cell), $\gamma_{q\nu}$ the linewidth of the phonon mode $\nu$ at the wave vector $q$, and $N_{q\nu}$ the total number of $q\nu$ points in the sum.

The critical superconductor temperature, $T_c$, was estimated with three different formulas: the McMillan formula[55], $T_c^{McM}$, the Allen-Dynes formula[56], $T_c^{AD}$, and the modified Allen-Dynes formula[29], $T_c^{mAD}$:

$$T_c^{McM} = \frac{\omega_{log}}{1.20} \times \exp\left[-\frac{1.04(1+\lambda)}{\lambda - \mu^*(1+0.62\lambda)}\right], \qquad (3)$$

$$T_c^{AD} = f_1 f_2 T_c^{McM}, \qquad (4)$$

$$T_c^{mAD} = (1.0061 + 0.0663\lambda)T_c^{AD}, \qquad (5)$$

where $\mu^*$ is the Coulomb pseudopotential, for which we selected values within the widely accepted range of 0.10–0.13, and the parameters $f_1$

and $f_2$ are defined like:

$$f_1 = \left[1 + \left(\lambda/\Lambda_1\right)^{3/2}\right]^{1/3}$$

$$f_2 = 1 + \frac{\left(\bar{\omega}_2/\omega_{log} - 1\right)\lambda^2}{\lambda^2 + \Lambda_2^2}, \tag{6}$$

with $\bar{\omega}_2 = \langle\omega^2\rangle^{1/2}$, $\Lambda_1 = 2.46(1 + 3.8\mu^*)$ and $\Lambda_2 = 1.82(1 + 6.3\mu^*)(\bar{\omega}_2/\omega_{log})$. Meanwhile, the logarithmic average phonon frequency, $\omega_{log}$, is defined like:

$$\omega_{log} = \exp\left[\frac{2}{\lambda}\int_0^\infty \frac{d\omega}{\omega}\alpha^2 F(\omega)ln(\omega)\right], \tag{7}$$

and the EPC strength, $\lambda$, is proportional to the first inverse momentum of the spectral function, namely:

$$\lambda = 2\int_0^\infty \frac{d\omega}{\omega}\alpha^2 F(\omega) = \frac{1}{N_{q\nu}}\sum_{q\nu}\lambda_{q\nu}, \tag{8}$$

where

$$\lambda_{q\nu} = \frac{\gamma_{q\nu}}{\pi\hbar N(E_F)\omega_{q\nu}^2}. \tag{9}$$

The $\lambda_{qv}$ parameter in the equation above corresponds to the EPC strength of the phonon mode at wave vector $q$ and phonon branch $v$. All electron-phonon coupling and $T_c$ results are listed in Table S3.

To evaluate the changes to the excitonic binding energy arising from electron-phonon coupling, we first calculated the phonon eigenvalues and eigenvectors on a $4 \times 4 \times 4$ $\mathbf{q}$-point grid using the finite displacement method in conjunction with nondiagonal supercells[57]. We then used those phonons to generate atomic configurations in a $4 \times 4 \times 4$ supercell in which the atoms were displaced along thermal lines[32], providing representative configurations that the atoms adopt at zero temperature. We then calculated the average of the electronic properties of the system over these configurations to obtain an estimate of the changes to the band gap and excitonic binding energy driven by quantum fluctuations of the helium nuclei, as shown in Table S6. Due to the high phonon energies in solid helium, thermal fluctuations only become important at temperatures higher than those of interest in this work[8].

We also qualitatively evaluated the effects of electron-phonon coupling on the predicted superconductor transition temperature. For this test, we combined the electronic band structure obtained at 20 TPa (i.e., when the band gap closes in the absence EPC effects) with the phonon spectrum obtained at 30 TPa (i.e., when the band gap closes when considering EPC effects) to calculate the resulting Eliashberg function (Fig. S19). (These $\alpha^2 F$ calculations were performed at the PBE level, just like in the previous cases.) It was found that when EPC effects were approximately considered in such a manner, the predicted $T_c$, increased from 1 K (without corrections, Table S3) to 6 K.

## Data availability
The data that support the findings of this study are available upon request from C.L. and the corresponding authors.

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

## Acknowledgements

C.C. acknowledges support from the Spanish Ministry of Science, Innovation and Universities under the fellowship RYC2018-024947-I and grant TED2021-130265B-C22. This work has been also supported by the grant PID2020-113565GB-C21 funded by MCIN/AEI/10.13039/501100011033 and grant 2021 SGR 01411 from the Generalitat de Catalunya. C.L and C.C. thankfully acknowledge the computer resources at MareNostrum and the technical support provided by Barcelona Supercomputing Center (RES-FI-1-0006 and RES-FI-2022-2-0003). J.S. gratefully acknowledges financial support from the National Key R&D Program of China (grant nos. 2022YFA1403201), the National Natural Science Foundation of China (grant nos. 12125404, 11974162, and 11834006), and the Fundamental Research Funds for the Central Universities. Part of the calculations were carried out using supercomputers at the High Performance Computing Center of Collaborative Innovation Center of Advanced Microstructures, the high-performance supercomputing center of Nanjing University. L.J.L gratefully acknowledges the computational resources provided by the National Supercomputer Service through the United Kingdom Car-Parrinello Consortium (EP/P022561/1). I.E. and Y.-W.F. acknowledge funding from the European Research Council (ERC) under the European Union's Horizon 2020 research and innovation program (Grant Agreement No. 802533) and the Department of Education, Universities and Research of the Eusko Jaurlaritza and the University of the Basque Country UPV/EHU (Grant No. IT1527-22). C.L and C.C. acknowledge interesting discussions and kind assistance from Raymond C. Clay III on ultra-compressed helium pseudopotentials and from Jordi José on white dwarfs. B.M. acknowledges support from a UKRI Future Leaders Fellowship (Grant No. MR/V023926/1), from the Gianna Angelopoulos Programme for Science, Technology, and Innovation, and from the Winton Programme for the Physics of Sustainability. Part of the calculations were performed using resources provided by the Cambridge Tier-2 system (operated by the University of Cambridge Research Computing Service and funded by EPSRC [EP/P020259/1]).

## Author contributions

C.C. and J.B. conceived the study and planned the research. C.L. performed the DFT and EPC calculations and analysis. C.P. performed the AIRSS calculations and analysis. C.D., Q.L. and J.S. performed the all-electron and many-body perturbation GW calculations and analysis. Y.-W.F. and I.E. performed the SSCHA calculations and analysis. I.E., Y.-W.F., C.P., B.M., L.J.C., C.D. and J.S. crucially assisted on the electron-phonon coupling and superconducting critical temperature calculations. The manuscript was written by C.L. and C.C. with substantial input from the rest of co-authors.

## Competing interests

The authors declare no competing interests.
