## [Peer Review File · Nature Communications]

Excitonic Insulator to Superconductor Phase Transition in Ultra-Compressed HeliumREVIEWER COMMENTS

Reviewer #1 (Remarks to the Author):

The authors predict an excitonic insulator phase in hcp helium at high pressures above 20 TPa based on a suite of first-principles calculations for the electronic, structural, and vibrational properties of the solid. Furthermore, since the electron-phonon coupling is enhanced significantly at those high pressures, the metallic phase of hcp helium is predicted to be superconducting in addition. These results are very interesting and continue previous work on the behaviour of helium at extremely high pressures in the TPa range (see Refs. given in the introduction). They are of fundamental interest (phase diagram of helium at high pressures) and relevant for applications in astrophysics (e.g. white dwarfs) and for superconductivity. The manuscript is in general well written has the potential for being accepted in ncomms if the authors can address the following comments properly.

The discussion with respect to high pressures in astrophysics is a bit weak from my perspective. White Dwarfs are composed of a very dense electron-ion plasma which contains besides helium nuclei also carbon and oxygen nuclei as a result of the stellar nucleosynthesis along the main sequence. Furthermore, high pressures in the Gbar range well exceeding TPa's have been generated in the laboratory using the NIF as driver for strong shock waves. I recommend that the authors inspect results of the Discovery Science program of the NIF in order to include few relevant references. I also recommend to change the quote "10,000 GPa = 10 TPa" in the first paragraph by comparing the GPa and Mbar scales, e.g., "1 TPa = 10 Mbar".

The presented results are based on various ab initio calculations using DFT and many-body perturbation theory within the GW approximation. The authors have checked the performance of XC functionals and decided to use B3LYP because this functional predicts a band gap closure closest to DMC results (Ref. 1). Note that the value of 24 TPa given in the text is not correct. Inspecting figure 1a it should be rather 23 TPa.

The pressures for the band gap and the binding energy in Fig. 2d are different for B3LYP and GW. Please explain the differences in more detail. In the left column on page 4 a value of

“600 GPa” is mentioned which translated into 0.6 TPa – is this correct and if “yes” where is this behaviour shown? I also recommend to use TPa throughout the paper.

The EPC strength is estimated as described in the methods section. The authors state on p. 4 that $\alpha^2 F$ is “extremely high” – compared to what? Non-expert readers would be happy to get further guidance at that point since superconductivity at high pressures is currently a hot topic.

Many-body perturbation theory was applied for the calculation of the excitonic binding energy in the $G_0 W_0$ approximation, and the dielectric function was processed in the plasmon-pole approximation. What would be the effect of improving towards a full GW calculation and going beyond the plasmon-pole approximation? What would be the change of the imaginary parts of the self-energy and the dynamically screened potential? Would a further damping of the energies result? And thereby modify the results for the band structure in Fig. 2 which is the source for the identification of the excitonic insulator and metallic phase and of superconductivity as well? Please comment.

I also recommend to make a remark why the authors have used different XC functionals for different quantities. Is this critical for the results obtained? What would be the corresponding deviations with respect to B3LYP? Give at least estimates as long as test calculations have not been performed.

The inclusion of xenon in Fig. S10 for an experimental test is helpful as long as TPa pressures as necessary for helium can only be realized in shock-wave experiments which also generate much higher temperatures in the sample.

The high-pressure phase diagram and metallization of dense liquid helium has recently been addressed, see <https://doi.org/10.1103/PhysRevB.102.224107>

Typos: “tinny” on p. 2, “CBM minimum” on p. 3. “OTFG” on p.6 is not explained.

Reviewer #2 (Remarks to the Author):

This manuscript performs state-of-the-art numerical calculations of helium under extreme conditions of pressures to estimate how helium transitions from an insulator to a metal. The authors speculate that helium form a novel excitonic phase as the metallic density is reached. They also estimate the transition temperature below which one has a BCS superconductor. They suggest that similar effects could be seen in dense Xe. The research is very well supported and the manuscript is well written. I recommend publication.

Some comments:

- 1) What is the range of temperature of the white dwarf stars? The authors justify the calculations because of the application to white dwarfs but will WD be in a relevant range of temperatures for the excitonic or superconducting states to be present?
- 2) The authors comment that the hcp phase is very stable, however they also say that the c/a ratio varies. Hence the phase is not strictly speaking hcp (with an ideal c/a ratio).
- 3) why is figure 3d a sketch without units on the x or y axes? In the text they give definite predictions for the pressures, temperatures, binding energies etc. Can't they indicate their predictions on this figure?

Reviewer #3 (Remarks to the Author):

In the present manuscript, Liu et al. report on theoretical simulations of He compressed to ultra-high pressure. Using standard DFT methodology, the authors calculated electronic and vibrational properties of He, as well as electron-phonon coupling in the TPa pressure range. They found that helium becomes an excitonic insulator (EI) prior to reaching metallic state and predict that metallic helium should become a superconductor with a critical temperature of ≈ 30 K at 20 TPa and of ≈ 100 K at 100 TPa. While these findings are of interest for experts in certain branches of condensed matter physics, the results presented in the manuscript are not of sufficiently broad interest to justify publication in Nature Communications, and (at least in the present form) could be misleading for non-experts.

The main motivation for the broad interest of the obtained results is an importance of ultra-compressed helium for understanding of stars evolution in the Universe. However, while the authors indeed carry out simulations at extremely high pressure, up to 100 TPa, their main results, the EI and superconducting states of He are predicted to occur at temperatures 0-100K, and therefore unlikely to be relevant for properties of matter at truly extreme conditions. And an experimental verification of the theoretical predictions in the lab is not feasible. Thus, I would recommend to revise the manuscript and submit it to a more specialized journal.

Upon the revision, the authors should address some additional issues.

1. Electronic structure calculations used for the prediction of semiconductor-EI-metal transitions were carried out using Hybrid B3LYP functional chosen on the basis of a comparison with earlier diffusion Monte-Carlo (DMC) calculations in PRL 101, 106407 (2008), Ref. 1 in the manuscript. However, as the only criterion the authors use a comparison of the values of the band gap (Fig. 1). Such comparison has limited validity. At least equation of state could have been compared.

2. In PRL 112, 055504 (2014), Ref 2 of the present manuscript, some of the authors demonstrated that the effects of electron-phonon coupling on the band gap were substantial and determined the pressure for metal-insulator transition of solid He to be at 32.9 TPa at $T = 0$ K, which was significantly higher than 25.7 TPa in Ref. 1, used for the verification of the accuracy of the present calculations. I understand, of course, that similarly to Ref. 1, the present work employed the static lattice approximation. However, it is unclear how the effects of electron-phonon coupling would affect other (major) results at pressure below 32.9 TPa presented in the present manuscript.

3. The authors report their results as a function of pressure, likely determining pressure using different functionals, as the electronic structure (B3LYP), phonons, and electron-phonon couplings (PBE, PBEsol) were calculated in different approximations. One should read the manuscript carefully to understand, for instance, that the phonon and EPC calculations at 20 GPa have been likely performed in the metallic state, creating a mess for

non-experts as the manuscript also reports semiconductor and EI states at this pressure, depending on the approximation. In view of this, the authors report in the ABSTRACT T_c of 30 K at 20 TPa! Is the semiconductor state of He (below 25,7 or 32.9 TPa, see point 2) also a superconductor? Moreover, a minor modification of parameter μ in the calculations (which is just an "educated" guess anyhow) from 0.1 to 0.13 reduces the predicted T_c reported in the ABSTRACT by almost 50%!

3. A minor technical comment: it is difficult to reconcile the phonon dispersions in Fig. 3a and SFig. 5. Looking at the 20GPa curve, there is a distinct soft mode at the Λ point in SFig.5. However, it is not seen in Fig. 3a. The same holds for SFig.11c+d and SFig. 15b.

We would like to thank all the Reviewers for their thoughtful comments and useful suggestions on our manuscript. We are glad to acknowledge that most Reviewers who have evaluated our work recommend its publication after some revision.

Please, find below a point-by-point response to all the Reviewers' comments. The main changes accompanying this revision are highlighted in blue color in the resubmitted manuscript (along with a clean version of the full article). It is noted that we have thoroughly revised our article according to all the Reviewers' recommendations. For instance, we have performed additional many-body perturbation GW and electron-phonon coupling calculations, Figures 2 and 3 in the main text have been improved, and the Supplementary Information has been extended with 4 new figures and 2 new tables. At the end of this response letter, you will find a detailed list of the main changes applied to our manuscript.

We hope that the revised version of our article will be now accepted for publication in your journal.

Reviewer #1 (Remarks to the Author):

The authors predict an excitonic insulator phase in hcp helium at high pressures above 20 TPa based on a suite of first-principles calculations for the electronic, structural, and vibrational properties of the solid. Furthermore, since the electron-phonon coupling is enhanced significantly at those high pressures, the metallic phase of hcp helium is predicted to be superconducting in addition. These results are very interesting and continue previous work on the behaviour of helium at extremely high pressures in the TPa range (see Refs. given in the introduction). They are of fundamental interest (phase diagram of helium at high pressures) and relevant for applications in astrophysics (e.g. white dwarfs) and for superconductivity. The manuscript is in general well written has the potential for being accepted in ncomms if the authors can address the following comments properly.

Reply: We thank the Reviewer for his/her careful evaluation of our work and general positive evaluation.

1. The discussion with respect to high pressures in astrophysics is a bit weak from my perspective. White Dwarfs are composed of a very dense electron-ion plasma which contains besides helium nuclei also carbon and oxygen nuclei as a result of the stellar nucleosynthesis along the main sequence. Furthermore, high pressures in the Gbar range well exceeding TPa`s have been generated in the laboratory using the NIF as driver for strong shock waves. I recommend that the authors inspect results of the Discovery Science program of the NIF in order to include few relevant references. I also recommend to change the quote "10,000 GPa = 10 TPa" in the first paragraph by comparing the GPa and Mbar scales, e.g., "1 TPa = 10 Mbar".

Reply: Following the Reviewer's advice, we have revised and improved the motivation of our work in the context of astrophysics and ultra-high pressures. On one hand, we have better articulated the far-reaching consequences of our work as concerns the understanding of the physico-chemical evolution and cooling process of low-mass white dwarfs (WDs); for this end, we have added some additional explanatory sentences in the first paragraph of the Introduction as well as some relevant bibliographic references (i. e., [1]-[3]). On the other hand, we have mentioned the state-of-the-art experimental high-pressure work carried out at the National Ignition Facility and cited some related bibliographic references (i. e., [4]-[6]), also in the first paragraph of the Introduction. Finally, we have also quoted "1 TPa = 10 Mbar" in the Abstract (although we should note that we have stick to TPa as the default pressure unit throughout the manuscript).

2. The presented results are based on various ab initio calculations using DFT and many-body perturbation theory within the GW approximation. The authors have checked the performance of XC functionals and decided to use B3LYP because this functional predicts a band gap closure closest to DMC results (Ref. 1). Note that the value of 24 TPa given in the text is not correct. Inspecting figure 1a it should be rather 23 TPa.

Reply: We thank the Reviewer for having spotted this error, which have been corrected in the revised version of our manuscript.

3. The pressures for the band gap and the binding energy in Fig. 2d are different for B3LYP and GW. Please explain the differences in more detail. In the left column on page 4 a value of “600 GPa” is mentioned which translated into 0.6 TPa – is this correct and if “yes” where is this behaviour shown? I also recommend to use TPa throughout the paper.

Reply: Following the Reviewer’s recommendation, now we employ TPa as the default pressure unit throughout the manuscript. Regarding our excitonic insulator results, the DFT-B3LYP and many-body perturbation GW methods essentially provide different metallization pressures and different excitonic binding energies for solid ^4He . In particular, the DFT-B3LYP approach renders $|E_{\text{bind}}|=0.19$ eV close to a metallization pressure of ≈ 23 TPa and predicts a pressure interval of approximately 0.2 TPa in which a bulk excitonic insulator state should be stable. Meanwhile, the many-body perturbation GW approach, which for the calculation of electronic and excitonic properties should be considered as more accurate than DFT-B3LYP, renders $|E_{\text{bind}}|=0.34$ eV close to a metallization pressure of ≈ 23.4 TPa; consequently, the GW approach predicts a broader pressure interval of 0.45 TPa over which a bulk excitonic insulator state should be stable. (We should note that in our original submission the reported many-body perturbation theory results were obtained at the G_0W_0 level, thus were less accurate and slightly different from the new GW results reported in our revised manuscript -see the amended Methods section and also our response to point 5 below-.) In the revised Fig. 2d, the excitonic insulator results obtained with the DFT-B3LYP and GW approaches described above are presented simultaneously in a more clear and easy-to-compare format (Fig. R1 below). In addition, in column 2 in page 3 of the revised manuscript, we comment in more detail on these results.

Fig. R1. Comparison of the exciton binding energy and band gap calculated with B3LYP DFT (solid lines) and many-body perturbation theory within the full GW approximation (dashed lines). The cyan (B3LYP) and orange (GW) regions indicate the pressure range in which the sufficient condition for spontaneous formation of excitons is fulfilled.

4. The EPC strength is estimated as described in the methods section. The authors state on p. 4 that $\alpha^2 F$ is “extremely high” – compared to what? Non-expert readers would be happy to get further guidance at that point since superconductivity at high pressures is currently a hot topic.

Reply: In view of the Reviewer’s comment, in the revised version of our manuscript we now provide a physically meaningful comparison between the EPC strengths obtained in this study for solid ${}^4\text{He}$ at specific reciprocal space points and the average EPC strengths that have been reported for high-temperature superconductors like H_3S and LaH_{10} (the corresponding bibliographic references have been also added to the revised manuscript). In particular, in pages 4 and 5 now we write:

“The EPC strength associated to these phonon modes respectively amount to ≈ 140 and 40 (Tables S1-S2), which are extremely high as compared to the average λ of the high-temperature superconductors H_3S [30] and LaH_{10} [31] (i.e., $\sim 1-10$).”

5. Many-body perturbation theory was applied for the calculation of the excitonic binding energy in the G_0W_0 approximation, and the dielectric function was processed in the plasmon-pole approximation. What would be the effect of improving towards a full GW calculation and going beyond the plasmon-pole approximation? What would be the change of the imaginary parts of the self-energy and the dynamically screened potential? Would a further damping of the energies result? And thereby modify the results for the band structure in Fig. 2 which is the source for the identification of the excitonic insulator and metallic phase and of superconductivity as well? Please comment.

Reply: To answer the Reviewer’s questions, we have performed full many-body perturbation theory calculations within the GW approximation and replaced the previous excitonic binding energy results obtained at the G_0W_0 level by the new and more accurate GW results. In the revised manuscript, we show and explain our new GW results in the revised Fig.2d, column 2 in page 3, revised Methods section, new supplementary table Table S4, and the new supplementary figures Figs. S15-S16.

In particular, for a pressure of 23 TPa, we first tested the convergence of the energy band gap with respect to the number of frequency points (Fig. R1a below) and GW iterations (Fig. R1b below). We found that 800 frequency points and 8 GW iterations rendered our targeted band gap convergence threshold of < 0.1 eV. Subsequently, we calculated the GW band gap at several pressure points. It was found that in the full many-body perturbation GW calculations the band gap tends to open as compared to the results obtained at the G_0W_0 and DFT-PBE levels (Fig. R1c below). Finally, we calculated the excitonic energy map and excitonic binding energy for solid ${}^4\text{He}$ with the many-body perturbation GW method at several pressures points (Fig. R2 below), and compared them with the analogous DFT-B3LYP results. (See also our response to the previous point 3.)

Fig. R1. Many-body perturbation GW calculations of the band gap at a pressure of 23 TPa. Energy band gap convergence with respect to **a)** the number of frequency points and **b)** $G_n W_n$ iterations. **c)** Estimated band gaps at five different pressures with the DFT-PBE, $G_0 W_0$, and full GW methods.

Fig. R2. Pressure evolution of the excitonic energy obtained with full GW calculations. The interpolated exciton energies are mapped into different q-point with colorful curves indicating different bands. The exciton in path K- Γ has the lowest energy with a value of -0.34 eV at 23.25 TPa, which is negative, indicating that the exciton state is bound.

6. I also recommend to make a remark why the authors have used different XC functionals for different quantities. Is this critical for the results obtained? What would be the corresponding deviations with respect to B3LYP? Give at least estimates as long as test calculations have not been performed.

Reply: Due to several numerical accuracy and computational affordability reasons, in this work different exchange-correlation functionals have been employed in a well-reasoned manner for estimating different physical quantities. In Fig 1, we demonstrated through a series of benchmarking calculations that the hybrid B3LYP functional best reproduced the electronic properties of solid ^4He . Thus, we calculated the band structure and excitonic insulator properties of solid helium with this hybrid functional (Fig. 2). However, it is not computationally feasible to perform full phonon spectrum calculations considering large enough supercells (e. g., $4\times 4\times 4$) and electron-phonon coupling calculations considering dense q-point meshes (e. g., $16\times 16\times 8$) with the B3LYP functional. Consequently, for these two latter types of computationally very intensive calculations we employed the computationally more affordable, and still accurate, semi-local PBE functional. In the revised version of our manuscript, this information appears more clearly explained in the main text and figure captions.

In addition, motivated by the Reviewer's comment, we performed a simple test to assess the likely deviations resulting from using either hybrid or semi-local DFT approaches as concerns the estimation of phonon properties. In particular, we considered a relatively small $2\times 2\times 2$ supercell which, although it does not provide results converged to our desired level of accuracy, is computationally affordable within both the hybrid and semi-local DFT approaches. The results of our B3LYP and PBE phonon spectrum calculations employing such a reduced supercell are shown in the new supplementary figure Fig. S6, which is enclosed below. As it can be appreciated therein, the phonon discrepancies in both cases turn out to be pretty small and mostly negligible, made the only exception of the first optical eigenmode appearing at the Γ point. Therefore, it can be concluded that the likely deviations between the semi-local and hybrid DFT approaches as concerns the estimation of vibrational properties are overall pretty small. In the revised version of our manuscript, we comment on this point in page 3 (column 2).

Fig. R1. Phonon spectrum calculated with the semi-local PBE and hybrid B3LYP DFT functionals considering a relatively small $2\times 2\times 2$ supercell (which can be feasibly simulated in both cases).

7. The inclusion of xenon in Fig. S10 for an experimental test is helpful as long as TPa pressures as necessary for helium can only be realized in shock-wave experiments which also generate much higher temperatures in the sample.

Reply: We thank the Reviewer for his/her reassuring comment. We actually expect to motivate some exciting experiments on solid Xenon with our work.

8. The high-pressure phase diagram and metallization of dense liquid helium has recently been addressed, see <https://doi.org/10.1103/PhysRevB.102.224107>

Reply: We thank the Reviewer for drawing our attention on such an interesting theoretical work. In the revised version of our article, we have added a reference to that study in the Introduction.

9. Typos: "tinny" on p. 2, "CBM minimum" on p. 3. "OTFG" on p.6 is not explained.

Reply: We thank the Reviewer for having spotted such typos. Those have been corrected in the revised version of our manuscript.

Reviewer #2 (Remarks to the Author):

This manuscript performs state-of-the-art numerical calculations of helium under extreme conditions of pressures to estimate how helium transitions from an insulator to a metal. The authors speculate that helium form a novel excitonic phase as the metallic density is reached. They also estimate the transition temperature below which one has a BCS superconductor. They suggest that similar effects could be seen in dense Xe. The research is very well supported and the manuscript is well written. I recommend publication.

Reply: We thank the Reviewer for his/her careful evaluation of our work and very positive evaluation.

Some comments:

1) What is the range of temperature of the white dwarf stars? The authors justify the calculations because of the application to white dwarfs but will WD be in a relevant range of temperatures for the excitonic or superconducting states to be present?

Reply: Since white dwarfs (WDs) do not have an internal energy source, they cool down over time until eventually reaching a temperature of 2.7 K, which corresponds to the current Universe temperature due to the cosmic background radiation [Annu. Rev. of Astron. and Astrophys. **28**, 139 (1990)]. Understanding the cooling process of WDs is essential for inferring their physico-chemical evolution and also provide bounds for the age of the Universe. For low-mass WDs mostly containing solid helium, our discovery of superconductivity should have important consequences on their cooling rates (heat transport mechanisms and the generation of magnetic fields in the interior of WDs should be affected by our findings). In the revised version of our manuscript, we have better articulated the far-reaching consequences of our work as concerns the understanding of the physico-chemical evolution and cooling process of low-mass WDs; for this, we have added some additional explanatory sentences in the first paragraph of the Introduction as well as some relevant bibliographic references (i. e., [1]-[3]).

2) The authors comment that the hcp phase is very stable, however they also say that the c/a ratio varies. Hence the phase is not strictly speaking hcp (with an ideal c/a ratio).

Reply: The Reviewer is totally right in that solid helium in the TPa regime is not an ideal hcp structure since its c/a ratio significantly departs from the ideal value of 1.63 (Fig. S10). Nevertheless, since the space group ($P6_3/mmc$) and crystal symmetry of the crystal remains the same from zero pressure all the way up to 100 TPa, we keep labeling this structure as hcp throughout the manuscript. Actually, this is normal practice done in both theoretical and experimental high-pressure studies (see, for instance, Phys. Rev. B 83, 144114 -2011- and Appl. Phys. Lett. 92, 111911 -2008-). In view of the Reviewer's comment, in the revised version of our manuscript we have clarified this point in page 3 (column 1).

3) Why is figure 3d a sketch without units on the x or y axes? In the text they give definite predictions for the pressures, temperatures, binding energies etc. Can't they indicate their predictions on this figure?

Reply: Following the Reviewer's suggestions, in the revised version of our manuscript we have modified

Fig. 3d in order to provide some details on the order of magnitude of the temperatures and pressures involved in the excitonic insulator to superconductor and superconductor to semimetal phase transitions in solid helium (see below). We note in passing also that we have slightly modified the phase boundary that was originally drawn for delimiting the excitonic insulator and superconductor states.

Fig. R1. Qualitative sketch of the possible phase diagram of ultra-compressed solid helium based on previous results [Phys. Rev. B **74**, 165107] and the key physical findings presented in our work.

Reviewer #3 (Remarks to the Author):

In the present manuscript, Liu et al. report on theoretical simulations of He compressed to ultra-high pressure. Using standard DFT methodology, the authors calculated electronic and vibrational properties of He, as well as electron-phonon coupling in the TPa pressure range. They found that helium becomes an excitonic insulator (EI) prior to reaching metallic state and predict that metallic helium should become a superconductor with a critical temperature of ≈ 30 K at 20 TPa and of ≈ 100 K at 100 TPa. While these findings are of interest for experts in certain branches of condensed matter physics, the results presented in the manuscript are not of sufficiently broad interest to justify publication in Nature Communications, and (at least in the present form) could be misleading for non-experts.

The main motivation for the broad interest of the obtained results is an importance of ultra-compressed helium for understanding of stars evolution in the Universe. However, while the authors indeed carry out simulations at extremely high pressure, up to 100 TPa, their main results, the EI and superconducting states of He are predicted to occur at temperatures 0-100K, and therefore unlikely to be relevant for properties of matter at truly extreme conditions. And an experimental verification of the theoretical predictions in the lab is not feasible. Thus, I would recommend to revise the manuscript and submit it to a more specialized journal.

Reply: We thank the Reviewer for his/her thorough evaluation of our work. Nevertheless, we do not share his/her opinion on that *“the results presented in the manuscript are not of sufficiently broad interest to justify publication in Nature Communications”*.

The Reviewer argued that the temperatures in the range of 0-100K are not relevant for Astrophysics. This is actually not the case. At the end of their lives, white dwarfs (WDs) do not have an internal source of energy and consequently cool down over time eventually reaching a temperature of 2.7 K (i.e., the current temperature of the Universe due to cosmic background radiation). Understanding the cooling process of WDs is critical for inferring their evolution over time and thus provide physically well motivated bounds for the age of the Universe [see, for instance, the topic articles *“Cooling of White Dwarfs”*, Annu. Rev. of Astron. and Astrophys. 28, 139 (1990) and *“Current challenges in the physics of white dwarf stars”*, Phys. Rep. 988, 1 (2022)]. Actually, this is one of the main reasons why researchers drew their attention on metallic helium in first instance: insulator materials dissipate and conduct heat very differently from what metallic systems do (and metallic and superconductor materials are also very much different on this regard). The fact that solid helium becomes superconductor below a certain critical temperature, an information that prior to our work was not known, will inevitably affect the theoretical models and assumptions employed for estimating the cooling rate of white dwarfs, and will also lead to a rethinking of the interactions of WDs with magnetic fields.

Let us assure the validity of our argument above by literally quoting one paragraph extracted from the recent review article *“Current challenges in the physics of white dwarf stars”* by Saumon et al., which has been published in the highly reputed journal Physics Reports (Phys. Rep. 988, 1 -2022-) [Section 2.2]: *“Once a star reaches the white dwarf state, it simply cools down for the rest of its evolution. Accurately modeling this cooling process has been one of the central goals of white dwarf research for over half a century [23–27]. While the evolution of white dwarfs is relatively simple compared to that of other astrophysical objects, it demands a precise description of a variety physical processes, which take place under extreme conditions seldom found elsewhere in the Universe. The rate at which a white dwarf*

cools down depends on the neutrino production rate in its core, the thermal conductivity of its core and envelope, the radiative opacity of its atmosphere, element transport in its interior, and the physics of phase transitions. The problem of white dwarf cooling is therefore far from trivial, but the unique potential of white dwarf cosmochronology (Section 2.5) motivates theoretical and experimental efforts to improve our description of the physical processes at play.”

Therefore, it is plainly clear that our novel results on the excitonic insulator and superconductor behavior of solid helium at ultra-high pressures, and the involved phase transition, are of high interest and relevance to the broad community of Astrophysics (as well as to Condensed Matter Physics). In the revised version of our manuscript, we have better articulated the far-reaching consequences of our work as concerns the understanding of the physico-chemical evolution and cooling process of low-mass WDs; for this end, we have added some additional explanatory sentences in the first paragraph of the Introduction as well as some relevant bibliographic references (i. e., [1]-[3]).

Furthermore, the Reviewer states that “*experimental verification of the theoretical predictions in the lab is not feasible*”. It is certain that pressures of about 10-100 TPa currently are not attainable in experimental high-pressure facilities. Nevertheless, we may also challenge this view based on two facts. Firstly, just like another Reviewer has noted, ultra-high pressures of the order of several TPa have been already generated in the laboratory by using the National Ignition Facility (NIF) as a driver for strong shock waves (i.e., ~5 TPa). Thus, we are actually not that far from a possible experimental verification of our theoretical predictions on solid helium in the near future. (In the revised version of our manuscript, we have commented on this point and cited some NIF related bibliographic references, namely, [4]-[6].) Moreover, Aluminium has been already compressed to 400 TPa under shock-wave conditions generated by an underground nuclear explosion, although such experiments cannot be carried out under present agreements (see Nature Materials 9, 624 -2010- and references therein). And secondly, as it appears explained in the main text, Conclusions and Supplementary Information, we have also disclosed superconductivity in compressed solid xenon and demonstrated that this is fundamentally very similar to that of solid helium; our prediction for the critical superconductivity pressure of solid xenon is about 140 GPa (see, for instance, supplementary figure Fig. S12), a compression that is perfectly attainable in current high-pressure experimental facilities. Therefore, some of the results reported in this study for solid helium may actually be verified in the lab, although indirectly.

1. Electronic structure calculations used for the prediction of semiconductor-El-metal transitions were carried out using Hybrid B3LYP functional chosen on the basis of a comparison with earlier diffusion Monte-Carlo (DMC) calculations in PRL 101, 106407 (2008), Ref. 1 in the manuscript. However, as the only criterion the authors use a comparison of the values of the band gap (Fig. 1). Such comparison has limited validity. At least equation of state could have been compared.

Reply: We agree with the Reviewer in that benchmarking our theoretical DFT results for ultra-compressed solid helium only for the electronic band gap may seem a bit limited. However, to the best of our knowledge, no other physical quantities estimated with highly-accurate first-principles methods (e.g., quantum Monte Carlo approaches) have been reported in the literature for solid helium in the TPa regime. For instance, the equations of state reported in Fig.3 of the work mentioned by the Reviewer (PRL 101, 106407 -2008-) correspond to DFT calculations performed with semi-local DFT exchange correlation functionals (not to diffusion Monte Carlo calculations).

In view of the Reviewer's comment, and to possibly motivate future highly-accurate first-principles studies on solid helium in the TPa regime, in the revised version of our article we have added a supplementary figure in which we show the equations of state of ultra-compressed solid helium obtained with different semi-local and hybrid DFT functionals (supplementary figure Fig. S11, quoted below).

Fig. R1. Equations of state of hcp ^4He obtained with different DFT exchange-correlation functionals.

2. In PRL 112, 055504 (2014), Ref 2 of the present manuscript, some of the authors demonstrated that the effects of electron-phonon coupling on the band gap were substantial and determined the pressure for metal-insulator transition of solid He to be at 32.9 TPa at $T = 0$ K, which was significantly higher than 25.7 TPa in Ref. 1, used for the verification of the accuracy of the present calculations. I understand, of course, that similarly to Ref. 1, the present work employed the static lattice approximation. However, it is unclear how the effects of electron-phonon coupling would affect other (major) results at pressure below 32.9 TPa presented in the present manuscript.

Reply: This is actually a very interesting point. All the results presented in our study have been obtained within the static lattice approximation (i.e., ions remain frozen on their equilibrium positions at $T=0\text{K}$). However, as noted by the Reviewer, it has been previously shown that quantum dynamical effects on the atomic positions have a non-negligible impact on the electronic properties of ultra-compressed solid helium (e.g., the metallization pressure shifts from ≈ 26 TPa under the static assumption to ≈ 33 TPa upon consideration of quantum dynamical effects on the atomic positions). Therefore, a well-founded question is: how such type of quantum dynamical effects may affect the excitonic insulator and superconductor temperature results reported in our study?

To provide an exact quantitative answer to the question above is out of the scope of the present work since it would be necessary to develop first novel state-of-the-art simulation methodologies that most likely would not be computationally feasible at the moment (recall that, as one can realize from inspecting the Methods section and technical information in the Supplementary Information, our excitonic insulator and T_c first-principles calculations performed within the static approximation are already in the

limit of what can be technically done in practice). Nevertheless, we have been able to provide an approximate quantitative answer to the question above by proceeding as described next.

First, to evaluate the influence of dynamical effects on the excitonic binding energy, E_{bind} , we generated a number of supercell configurations for a pressure of 18 TPa in which the atoms were displaced along representative thermal lines determined with the method described in work Phys. Rev. B 93 014302 (2016). Subsequently, we calculated the excitonic binding energy for those configurations by using the Wannier-Mott model, which we have demonstrated to do a good job for solid helium, along with the band gap (E_g), and computed their average values (Methods and Supplementary Information and Table R1 below). It was found that quantum dynamical effects on the atomic positions (1) significantly open the band gap, in agreement with previous DFT calculations (Phys. Rev. Lett. 112, 055504 -2014-), and (2) enhance the excitonic binding energy very substantially by >40% (i.e., $|E_{\text{bind}}|$ increases from 0.192 to 0.271 eV, see Methods, Supplementary Information and Table R1 below).

And second, we carried out a perturbative T_c calculation in which we combined the phonon spectrum obtained at ≈ 30 TPa, that is, the pressure at which solid helium becomes metallic when taking into consideration quantum dynamical effects, and the electronic band structure obtained at ≈ 20 TPa by neglecting quantum dynamical effects, which renders an incipient metallic state (Methods and Supplementary Information). (The assumption underlying such a procedure is that the specific electronic mechanisms governing the closure of the band gap in solid helium -e.g., the orbitals that overlap and their particular band topology- are barely influenced by quantum dynamical effects.) By proceeding so, we obtained a corrected T_c for a nominal pressure of ≈ 30 TPa that was six times higher than the corresponding value obtained when disregarding dynamical effects (i.e., T_c increased from ≈ 1 to 6 K, see Methods, Supplementary Information and Fig. R1 below).

Therefore, we may conclude that quantum dynamical effects on the atomic positions, in addition to displacing the metallization pressure of solid helium towards higher values, are also likely to enhance the excitonic binding energies and superconductor transition temperatures reported in this study. Thus, our results obtained within the static lattice approximation are physically sound and solid. In the revised version of our manuscript, we have described these new calculations and results in the main text, Methods section and Supplementary Information.

Configuration #	$ E_{\text{bind}} $ (eV)	E_g (eV)
1	0.256	1.315
2	0.215	1.408
3	0.213	1.266
4	0.253	1.475
5	0.299	1.192
6	0.232	1.157
7	0.258	1.220
8	0.420	1.407
9	0.384	1.482
10	0.183	1.511
Dynamical (with EPC effects)	0.271	1.343
Static (without EPC effects)	0.192	0.316

Table. R1. Band gap and excitonic binding energy obtained for hcp helium by considering and disregarding quantum dynamical effects on the atomic positions (EPC effects).

Fig. R1. Eliashberg spectral functions obtained at 20 TPa and 30 TPa by disregarding quantum dynamical effects on the atomic positions and by considering them (EPC) at 30 TPa.

3. The authors report their results as a function of pressure, likely determining pressure using different functionals, as the electronic structure (B3LYP), phonons, and electron-phonon couplings (PBE, PBEsol) were calculated in different approximations. One should read the manuscript carefully to understand, for instance, that the phonon and EPC calculations at 20 GPa have been likely performed in the metallic state, creating a mess for non-experts as the manuscript also reports semiconductor and EI states at this pressure, depending on the approximation. In view of this, the authors report in the ABSTRACT T_c of 30 K at 20 TPa! Is the semiconductor state of He (below 25,7 or 32.9 TPa, see point 2) also a superconductor? Moreover, a minor modification of parameter λ in the calculations (which is just an "educated" guess anyhow) from 0.1 to 0.13 reduces the predicted T_c reported in the ABSTRACT by almost 50%!

Reply: Due to multiple factors related to numerical accuracy and computational cost, in this work different exchange-correlation functionals have been employed in a well-reasoned manner for estimating different physical quantities. In Fig 1, we demonstrated through a series of benchmarking calculations that the hybrid B3LYP functional best reproduced the electronic properties of solid ^4He . Thus, we calculated the band structure and excitonic insulator properties of solid helium with this hybrid functional (Fig. 2). However, it is not computationally feasible to perform full phonon spectrum calculations considering large enough supercells (e. g., $4 \times 4 \times 4$) and electron-phonon coupling calculations considering dense q-point meshes (e. g., $16 \times 16 \times 8$) with the B3LYP functional. Consequently, for these

two latter types of computationally very intensive calculations we employed the computationally more affordable, and still accurate, semi-local PBE functional. In the revised version of our manuscript, this information appears more clearly explained in the main text and figure captions.

Regarding the effect of the selected Coulomb pseudopotential value, μ^* , on the reported T_c 's, it is worth noting that (1) this kind of arbitrariness is present in every single first-principles superconductivity study, and (2) $\mu^*=0.10$, the value that we have selected for reporting T_c values in the Abstract and main text, is the typical value adopted for metals in most published theoretical works. Nevertheless, at the same time we should acknowledge that the Reviewer's criticism is not out of reason. In view of the Reviewer's comment, and for cautious reasons, in the revised version of our article now all the T_c values reported in the Abstract and main text correspond to those calculated with $\mu^*=0.13$, which probably provide a lower bound of the true superconductor critical temperatures. In any case, the main conclusions reported in our study are not affected by such numerical nuances.

4. A minor technical comment: it is difficult to reconcile the phonon dispersions in Fig. 3a and SFig. 5. Looking at the 20GPa curve, there is a distinct soft mode at the Λ point in SFig.5. However, it is not seen in Fig. 3a. The same holds for SFig.11c+d and SFig. 15b.

Reply: The Reviewer is certainly right. The reason for the spotted differences is that different semi-local DFT functionals were employed for the calculation of the phonon spectra enclosed in Fig. 3 and Fig. S5 (in particular, the PBE functional was used in the first case and the PBEsol functional in the latter case). For the same reasons that different DFT functionals may provide different metallization pressures (Fig. 1 in the main text), different DFT functionals may also provide different vibrational features. Nevertheless, all the EPC and T_c results reported in the main text have been systematically and carefully obtained with the PBE functional. The results in the Supplementary Information mentioned by the Reviewer were obtained with the PBEsol functional and conform few numerical tests from the many undertaken. In any case, the semi-local PBE and PBEsol functionals provide overall equivalent results in terms of electronic band structure and vibrational properties for ultra-compressed solid helium, as we have mentioned in the main text and demonstrated in the Supplementary Information. In the revised version of our manuscript, we have been very careful in clearly quoting which DFT functional has been used in each particular calculation.

List of main changes applied on the manuscript:

1. Highly accurate many-body perturbation calculations within the GW approximation have been performed and reported in Figure 2 of the main text, as well as in the supplementary figures Figs. S15 and S16 and the supplementary table Table S4.
2. Additional electron-phonon coupling (EPC) calculations have been performed in order to assess the influence of dynamical EPC effects on the reported excitonic binding energies and superconductor critical temperatures, which originally were obtained by disregarding dynamical effects on the atomic positions (main text, Methods and supplementary table Table S6).
3. Complementary phonon calculations considering a relatively small 2x2x2 supercell have been performed with the hybrid B3LYP functional (see the supplementary figure Fig. S6).
4. The equations of state obtained with the most representative DFT functionals considered in this study are now provided in the supplementary figure Fig. S11 and are discussed in the main text.
5. Two out of the three figures appearing in the main text have been revised and improved according to the Reviewers' comments.
6. The Supplementary Information now contains 4 additional supplementary figures and 2 additional supplementary tables.
7. Up to 11 new bibliographic references have been added to the main text in order to better contextualize the motivation of our work and improve the analysis of our results.

REVIEWER COMMENTS

Reviewer #1 (Remarks to the Author):

The authors have carefully revised their manuscript according to the questions and remarks of the referee(s), in particular with respect to the performance of the AIMD simulations, the use of different XC functionals, and the accuracy of the GW approximation. The additional information given in the SM is very helpful for a more detailed insight into the calculations. I recommend publication of the article.

Remarks:

1. The second sentence in the introduction should be removed (or revised). WDs are final states of stars with core mass less than the Chandrasekhar limit - fusion reactions no longer occur. And they are not mono-elemental - C and O nuclei (and others) are also present.
2. Fig. S10 inset: It is interesting that the QE shows a feature in the c/a ratio in the range of the band-gap closure at 25 TPa while VASP yields a smooth curve - any explanation for the disagreement?

Reviewer #2 (Remarks to the Author):

I think that the author's responses to the referee's reports are satisfactory and the manuscript is ready for publication without further changes.

Reviewer #3 (Remarks to the Author):

The resubmitted manuscript still contains clearly incorrect statements and conclusions not supported with data. For example, the claim in the abstract that at 20 TPa helium is metallic superconductor with a critical temperature of ≈ 20 K is certainly incorrect. It is an artifact of the use of a less accurate approximation within DFT simulations and contradicts other data presented in the manuscript, e.g. in Fig. 2, where metallization pressure calculated using more accurate approximations is around 23 TPa. The authors do not report any estimation

of the metallization pressure in solid He including effects of electron-phonon interactions, but they admit that they influence the band gap. Most likely their inclusion would shift the metallization pressure to above 30 GPa. In obvious contradiction with the claim from the abstract of superconductivity of He at 20 GPa .

In their response the authors admit that they do not have adequate theoretical tools to consistently and quantitatively describe the phenomena that they study. They may have captured the picture qualitatively, but a study of superconductivity at $T \sim 10$ K and pressure above 20 TPa experimentally in a reasonably near future is highly unlikely, so no experimental verification/corrections of the claims presented in the manuscript can be expected. Therefore, the author's strategy of mixing different theoretical methods creates frustration and high risks to mislead the non-expert readers with quantitative numerical data, as in the example above. If Nature Communications considers this to be acceptable, the paper can be published. However, I cannot recommend the publication.

Reviewer #1 (Remarks to the Author):

The authors have carefully revised their manuscript according to the questions and remarks of the referee(s), in particular with respect to the performance of the AIMD simulations, the use of different XC functionals, and the accuracy of the GW approximation. The additional information given in the SM is very helpful for a more detailed insight into the calculations. I recommend publication of the article.

A: We thank very much the Reviewer for taking his/her time in revising our manuscript and making a series of very insightful and useful comments on it. We also thank the Reviewer for his/her final positive recommendation on our work.

Remarks:

1. The second sentence in the introduction should be removed (or revised). WDs are final states of stars with core mass less than the Chandrasekhar limit - fusion reactions no longer occur. And they are not mono-elemental - C and O nuclei (and others) are also present.

A: Following the Reviewer's recommendation, in the revised version of our manuscript the first and second sentence in the Introduction has been revised according to his/her comment. In particular, now we write:

"In their final evolution stage, the majority of stars in the Universe become white dwarfs (WDs) mostly consisting of a mixture of helium, carbon, and oxygen atoms immersed in a sea of electrons. The lower the mass of the WD, the larger the relative abundance of helium."

2. Fig. S10 inset: It is interesting that the QE shows a feature in the c/a ratio in the range of the band-gap closure at 25 TPa while VASP yields a smooth curve - any explanation for the disagreement?

A: We do not have a definitive answer to this question. It is quite likely that the reason for the slight c/a ratio disagreement between the VASP and QE calculations is originated by the use of different pseudopotentials (i.e., "projector augmented-wave" in the former case and "ultrasoft" in the latter). Nevertheless, as it has been already emphasized in the main text, the agreement between the equations of state obtained with both DFT codes is extremely good. In the revised version of our manuscript, we have added the following informative sentence to the caption of Fig. S10:

"VASP calculations were performed with projector augmented-wave pseudopotentials while QE calculations were performed with ultrasoft pseudopotentials."

Reviewer #2 (Remarks to the Author):

I think that the author's responses to the referee's reports are satisfactory and the manuscript is ready for publication without further changes.

A: We thank very much the Reviewer for taking his/her time in revising our manuscript and making a series of very insightful and useful comments on it. We also thank the Reviewer for his/her final positive recommendation on our work.

Reviewer #3 (Remarks to the Author):

The resubmitted manuscript still contains clearly incorrect statements and conclusions not supported with data. For example, the claim in the abstract that at 20 TPa helium is metallic superconductor with a critical temperature of ≈ 20 K is certainly incorrect. It is an artifact of the use of a less accurate approximation within DFT simulations and contradicts other data presented in the manuscript, e.g. in Fig. 2, where metallization pressure calculated using more accurate approximations is around 23 TPa. The authors do not report any estimation of the metallization pressure in solid He including effects of electron-phonon interactions, but they admit that they influence the band gap. Most likely their inclusion would shift the metallization pressure to above 30 GPa. In obvious contradiction with the claim from the abstract of superconductivity of He at 20 GPa.

A: The value of the metallization pressure estimated for solid helium in fact depends on the employed computational method. In the Abstract, for the sake of consistency, we report the metallization pressure estimated with the semi-local PBE exchange-correlation functional because that is the method that we have used for the calculation of the superconducting critical temperature in solid ^4He , T_c . As it already appears explained in the main text, and we argued in the previous revision round, the T_c calculations presented in our work currently are computationally feasible only at the semi-local PBE level or equivalent. On the other hand, this type of information, which clearly appears in the body of the main text (see, for instance, Figure 3 caption, page 4 and the Methods section), is too technical to appear elaborated already in the Abstract.

Nevertheless, in view of the Reviewer's comment and to avoid any possible misunderstanding, in the revised Abstract now we do not explicitly quote the value of the metallization pressure estimated with the semi-local PBE exchange-correlation functional for solid ^4He . Specifically, we write:

"Accordingly, we predict metallic helium to be a superconductor with a critical temperature of ≈ 20 K just above its metallization pressure and of ≈ 70 K at 100 TPa."

In their response the authors admit that they do not have adequate theoretical tools to consistently and quantitatively describe the phenomena that they study. They may have captured the picture qualitatively, but a study of superconductivity at $T \sim 10$ K and pressure above 20 TPa experimentally in a reasonably near future is highly unlikely, so no experimental verification/corrections of the claims presented in the manuscript can be expected. Therefore, the author's strategy of mixing different theoretical methods creates frustration and high risks to mislead the non-expert readers with quantitative numerical data, as in the example above. If Nature Communications considers this to be acceptable, the paper can be published. However, I cannot recommend the publication.

A: The main Reviewer's criticism on our work, namely, "Therefore, the author's strategy of mixing different theoretical methods creates frustration and high risks to mislead the non-expert readers with quantitative numerical data, as in the example above.", appears to be unjustified. First, by reporting the numerical results obtained with different first-principles methods of diverse precision and computational affordability, most likely we are not causing any frustration; rather, we are providing a comprehensive and well reasoned picture of what can be achieved with currently available *ab initio* methods, ranging

from standard to state-of-the-art techniques in terms of accuracy. And second, in the “Aims & Scope” section of the Nature Communications journal it appears written:

*“Nature Communications is an open access, multidisciplinary journal dedicated to publishing high-quality research in all areas of the biological, health, physical, chemical, Earth, social, mathematical, applied, and engineering sciences. **Papers published by the journal aim to represent important advances of significance to specialists within each field.**”*

Therefore, the Reviewer’s main concern on the hypothetically possible issues generated by our study on non-expert readers, does not seem to align with the general publication policy of Nature Communications.

REVIEWERS' COMMENTS

Reviewer #3 (Remarks to the Author):

In the resubmitted manuscript the authors have found a way to reduce risks to mislead scientists from other fields. I still think that a report based on a mixture of different theoretical methods that could create frustration is more suitable for a specialized journal. However, the revised manuscript could also be published in Nature Communications.

Reviewer #3 (Remarks to the Author):

In the resubmitted manuscript the authors have found a way to reduce risks to mislead scientists from other fields. I still think that a report based on a mixture of different theoretical methods that could create frustration is more suitable for a specialized journal. However, the revised manuscript could also be published in Nature Communications.

A: We thank the Reviewer for his careful evaluation of our work and final positive recommendation on it.